# Effect of Isoscopoletin on Cytokine Expression in HaCaT Keratinocytes and RBL-2H3 Basophils: Preliminary Study

**DOI:** 10.3390/ijms25136908

**Published:** 2024-06-24

**Authors:** Da-Yun Seo, Ji-Won Park, Seung-Ho Kim, Sei-Ryang Oh, Sang-Bae Han, Ok-Kyoung Kwon, Kyung-Seop Ahn

**Affiliations:** 1College of Pharmacy, Chungbuk National University, Cheongju 28160, Republic of Korea; sdy45120@kribb.re.kr (D.-Y.S.); shan@chungbuk.ac.kr (S.-B.H.); 2Natural Medicine Research Center, Korea Research Institute of Bioscience and Biotechnology, Cheongju 28116, Republic of Korea; kshkorea1@kribb.re.kr (S.-H.K.); seiryang@kribb.re.kr (S.-R.O.); 3Practical Research Division, Honam National Institute of Biological Resources (HNIBR), Mokpo 58762, Republic of Korea; jjiwon2@hnibr.re.kr

**Keywords:** isoscopoletin, atopic dermatitis, inflammation, keratinocytes, basophils, HaCaT, RBL-2H3

## Abstract

Isoscopoletin is a compound derived from various plants traditionally used for the treatment of skin diseases. However, there have been no reported therapeutic effects of isoscopoletin on atopic dermatitis (AD). AD is a chronic inflammatory skin disease, and commonly used treatments have side effects; thus, there is a need to identify potential natural candidate substances. In this study, we aimed to investigate whether isoscopoletin regulates the inflammatory mediators associated with AD in TNF-α/IFN-γ-treated HaCaT cells and PMA/ionomycin treated RBL-2H3 cells. We determined the influence of isoscopoletin on cell viability through an MTT assay and investigated the production of inflammatory mediators using ELISA and RT-qPCR. Moreover, we analyzed the transcription factors that regulate inflammatory mediators using Western blots and ICC. The results showed that isoscopoletin did not affect cell viability below 40 μM in either HaCaT or RBL-2H3 cells. Isoscopoletin suppressed the production of TARC/CCL17, MDC/CCL22, MCP-1/CCL2, IL-8/CXCL8, and IL-1β in TNF-α/IFN-γ-treated HaCaT cells and IL-4 in PMA/ionomycin-treated RBL-2H3 cells. Furthermore, in TNF-α/IFN-γ-treated HaCaT cells, the phosphorylation of signaling pathways, including MAPK, NF-κB, STAT, and AKT/PKB, increased but was decreased by isoscopoletin. In PMA/ionomycin-treated RBL-2H3 cells, the activation of signaling pathways including PKC, MAPK, and AP-1 increased but was decreased by isoscopoletin. In summary, isoscopoletin reduced the production of inflammatory mediators by regulating upstream transcription factors in TNF-α/IFN-γ-treated HaCaT cells and PMA/ionomycin-treated RBL-2H3 cells. Therefore, we suggest that isoscopoletin has the potential for a therapeutic effect, particularly in skin inflammatory diseases such as AD, by targeting keratinocytes and basophils.

## 1. Introduction

Atopy is a hypersensitivity reaction to substances that are generally harmless, and when atopy occurs in the skin, it is called atopic dermatitis (AD) [1]. AD is defined as a chronic inflammatory skin disease caused by a complex interaction of environmental and genetic factors [2]. AD is particularly common in infancy and childhood and is accompanied by xerosis and pruritus. Erythema is present in acute lesions and lichenification occurs in chronic lesions [3,4]. This can cause depression due to external skin changes and other complications, such as allergic asthma and rhinitis, may occur [5]. The prevalence of AD is increasing over time, and the cause and best treatment of AD remain unclear. AD is a chronic disease that requires both treatment and management [2,5].

The pathogenesis of AD involves inflammatory and allergic reactions that occur in the skin (Figure 1). Keratinocytes, which constitute most of the epidermis, play a central role in the occurrence of inflammatory skin diseases [6]. Activated keratinocytes release cytokines and chemokines, and thymus- and activation-regulated chemokines (TARC/CCL17) and macrophage-derived chemokines (MDC/CCL22) are highly increased at the serum level in individuals with AD [7,8,9,10,11]. TARC and MDC bind to the C-C chemokine receptor type 4 (CCR4) and trigger the infiltration of T helper type 2 (Th2) cells, T helper type 17 (Th17) cells, T helper type 22 (Th22) cells, and basophils into the tissue [12,13].

Tumor necrosis factor-alpha (TNF-α), one of the inducers of keratinocytes, is a representative pro-inflammatory cytokine involved in the production of inflammatory mediators, including TARC and MDC, through nuclear factor kappa-light-chain-enhancer of activated B cells (NF-κB) and mitogen-activated protein kinase (MAPK) cascades [14,15]. Interferon-gamma (IFN-γ) is one of the most potent keratinocyte-activating factors expressed in T helper type 1 (Th1) cells [16,17]. IFN-γ also induces apoptosis and cleaves epithelial cadherin (E-cadherin), an intercellular adhesion protein, resulting in spongiform morphology in the epidermis. This morphology is regarded as a hallmark of eczematous lesions in AD [6]. Additionally, IFN-γ triggers the production of cytokines and chemokines through the Janus kinase/signal transducer and activator of transcription protein (JAK/STAT) and phosphoinositide 3-kinase/protein kinase B (PI3K/AKT) pathways in keratinocytes [18]. Therefore, the suppression of keratinocyte activation and apoptosis are potential targets for the treatment of AD [6].

Basophils, which are involved in allergic responses, are cells with granules containing various soluble mediators. Basophils originate from bone marrow, circulate in the blood, and infiltrate tissue when inflammation occurs [19,20]. Basophils are activated and degranulated by cytokines released from Th2 cells or immunoglobulin E (IgE) from B cells, and they release cytokines such as interleukin-4 (IL-4) and interleukin-13 (IL-13) [21,22,23,24,25]. These cytokines not only activate Th2 cells but also stimulate sensory neurons to cause itching, which damages the skin barrier and makes it easier for antigens and allergens to enter [26]. The accumulation of basophils and high levels of IL-4 are characteristic of AD skin [27].

As an inducer of basophils, phorbol 12-myristate 13-acetate (PMA) is a mitogen involved in various biological phenomena by directly binding to protein kinase C (PKC) at the cellular level [28,29,30]. Ionomycin is an ionophore that destroys intercellular concentration gradients by transferring divalent cations through lipophilic membranes [31,32]. It has been reported that when basophils are activated by PMA and ionomycin, the production of Th2-type cytokines such as IL-4 is induced through MAPK and activator protein-1 (AP-1), which are downstream signaling pathways activated following PKC phosphorylation [33].

Corticosteroids, which have been widely used to treat AD, are prescribed for short-term use for inflammatory reactions, but when used chronically, they have side effects such as sepsis, venous thromboembolism (VTE), and fracture. Furthermore, young patients chronically exposed to corticosteroids are at risk of immune dysregulation and developmental disability [19,34,35]. Dupilumab, an IL-4 inhibitor, is expensive and has side effects such as conjunctivitis [36]. Natural-product-derived compounds might have relatively high bioavailability and low toxicity, so they can sometimes be used as safe candidates without side effects.

Isoscopoletin is isolated from ethyl acetate extracts or the acetate fraction of extracts from various plants with confirmed pharmacological activity. It has anti-hepatotoxic activity as a chemical component of preparations from *Fraxinus* species, and anti-bacterial activity as an extract of *Ranunculus japonicus* [37,38]. In addition, isoscopoletin is present in extracts from plants such as *Dryopteris fragrans* and *Ammi majus*, which have traditionally been used as treatments for skin diseases such as psoriasis [39,40]. Isoscopoletin is a coumarin derivative that is a secondary metabolite of plants, although there is limited research regarding its activity compared to its isomer, scopoletin [41,42]. Based on these pharmacological activities, we evaluated the potential of isoscopoletin to inhibit skin inflammation, which has not been reported to date.

Therefore, the purpose of this study was to identify the effects and mechanisms of isoscopoletin on the inflammatory response in HaCaT keratinocytes and RBL-2H3 basophils using biomarkers that can identify allergic disease. This study found that isoscopoletin has anti-inflammatory and anti-allergic effects through the regulation of multiple inflammatory pathways in HaCaT and RBL-2H3 cells, therefore, isoscopoletin is a candidate for AD treatment.

## 2. Results

### 2.1. Effect of Isoscopoletin on the Viability of HaCaT and RBL-2H3 Cells

To evaluate the physiological activity of isoscopoletin, first, the range that does not affect cell viability was determined. This range was investigated in HaCaT and RBL-2H3 cells via MTT assay. In the HaCaT cells, the cell viability was 96.49 ± 0.29% at 80 μM and 89.87 ± 0.01% at 160 μM, showing a significant decrease compared to the control group that was not treated with isoscopoletin (Figure 2A). In the RBL-2H3 cells, the cell viability was 91.82 ± 1.03% at 80 μM and 85.19 ± 0.44% at 160 μM, showing a significant decrease compared to the control group that was not treated with isoscopoletin (Figure 2B). Both cell lines showed no significant difference at concentrations below 40 μM. Therefore, isoscopoletin was applied in the following experiments at a concentration of 5–40 μM.

### 2.2. Isoscopoletin Inhibits Cytokine and Chemokine Release by HaCaT and RBL-2H3 Cells

Keratinocytes can be activated when pro-inflammatory cytokines, such as TNF-α and IFN-γ, are released into AD-afflicted skin [6]. Activated keratinocytes produce cytokines (IL-1β) and chemokines (TARC, MDC, MCP-1, and IL-8), and in the case of chemokines, the inflammatory response in the skin layer is intensified by inducing immune cell infiltration [6]. In accordance with the previous theory, ELISA confirmed that the protein levels of cytokines and chemokines were significantly increased in the TI-treated HaCaT cells and were decreased by isoscopoletin. At the highest concentration of 40 μM, isoscopoletin inhibited these levels by 62.11 ± 1.79% for TARC, 49.50 ± 2.95% for MDC, 30.77 ± 2.49% for MCP-1, 43.50 ± 4.77% for IL-8, and 38.11 ± 2.66% for IL-1β, compared to the only-TI-stimulated control group (Figure 3A–E).

Basophils are activated by PMA, which binds to PKC and ionomycin and disrupts the cellular concentration gradient. Activated basophils release IL-4 and stimulate sensory neurons, causing itching [4]. An ELISA confirmed that the protein level of IL-4 was significantly increased in PI-treated RBL-2H3 cells and was decreased by isoscopoletin. At the highest concentration of 40 μM, isoscopoletin inhibited IL-4 levels by 57.88 ± 3.80% compared to the only-PI-stimulated control group (Figure 3F).

### 2.3. Isoscopoletin Inhibits Gene Expression Levels of Cytokines and Chemokines in HaCaT Cells

The release of cytokines and chemokines confirmed in the previous results was additionally investigated at the mRNA transcription level using RT-qPCR. The mRNA levels of the cytokines and chemokines were significantly increased in the TI-treated HaCaT cells and were decreased by isoscopoletin. At the highest concentration of 40 μM, isoscopoletin inhibited these levels by 74.12 ± 7.54% for TARC, 52.16 ± 0.29% for MDC, 39.63 ± 0.83% for MCP-1, 65.06 ± 5.81% for IL-8, and 44.89 ± 7.71% for IL-1β, compared to the only-TI-stimulated control group (Figure 4A–E).

### 2.4. Isoscopoletin Suppresses Phosphorylation of MAPK and NF-κB Signaling Molecules in HaCaT Cells

In the TI-treated HaCaT cells, isoscopoletin decreased the protein and mRNA levels of cytokines and chemokines. Therefore, we analyzed the transcription factors regulated by TI using Western blot. MAPK is activated by various external stimuli and directs cellular responses. NF-κB is activated by the phosphorylation of IκBα, leading to translocation into the nucleus and the induction of inflammatory factors through mRNA transcription [17]. Thus, we investigated whether isoscopoletin could reduce the TI-induced phosphorylation of MAPK subunits (ERK, JNK, and p38), IκBα, and NF-κB subunit p65. Western blot analysis confirmed that the phosphorylation of MAPK (ERK, JNK, and p38), IκBα, and NF-κB p65 was significantly increased in the TI-treated HaCaT cells and was decreased by isoscopoletin. At the highest concentration of 40 μM, isoscopoletin suppressed the phosphorylation of signaling molecules by 40.34 ± 1.61% for p-ERK, 46.04 ± 2.25% for p-JNK, 92.05 ± 2.25% for p-p38, 25.64 ± 1.86% for p-p65, and 52.42 ± 4.71% for p-IκBα compared to the only-TI-stimulated control group (Figure 5A,B). In addition, in the analysis of the nuclear translocation of NF-κB p65 by ICC, isoscopoletin showed an inhibitory effect on the translocation level of NF-κB p65 from the cytoplasm into the nucleus (Figure 5C).

### 2.5. Isoscopoletin Suppresses Phosphorylation of STAT and AKT Signaling Molecules in HaCaT Cells

IFN-γ activates the JAK-STAT and PI3K-AKT signaling pathways in HaCaT cells, leading to the activation of downstream cascades. When IFN-γ binds to its receptor on the cell surface, the JAK-STAT pathway has a role as a transcription factor through a cascade of phosphorylation. Ultimately, STAT is translocated into the nucleus and regulates gene expression. In addition, PI3K, an enzyme involved in various cellular functions, phosphorylates AKT and induces the protein synthesis pathway [15]. Because the activation of these signaling pathways causes inflammation, we investigated whether isoscopoletin could reduce the phosphorylation of STAT family members (STAT1 and STAT3) and AKT. A Western blot analysis confirmed that the phosphorylation of STAT1, STAT3, and AKT was significantly increased in the TI-treated HaCaT cells and was decreased by isoscopoletin. At the highest concentration of 40 μM, isoscopoletin inhibited signaling molecules by 20.52 ± 0.28% for p-STAT1, 44.26 ± 0.28% for p-STAT3, and 37.95 ± 3.67% for p-AKT, compared to the only-TI-stimulated control group (Figure 6A,B).

### 2.6. Isoscopoletin Suppresses Activation of PKC, MAPK, and AP-1 Signaling Molecules in RBL-2H3 Cells

PI activates the PKC pathway in RBL-2H3 cells, and PKC induces degranulation and the release of lipid mediators and cytokines. These mediators contribute to the initiation of allergic inflammatory responses [32,33]. Therefore, we evaluated whether isoscopoletin’s reduction of IL-4 led to the inhibition of PKC, MAPK subunits (ERK, JNK, and p38), and AP-1 subunit (c-Fos and c-Jun) activation. A Western blot analysis confirmed that the activation of PKC, MAPK (JNK), and AP-1 (c-Fos and c-Jun) was significantly increased in the PI-treated RBL-2H3 cells and was decreased by isoscopoletin. At the highest concentration of 40 μM, isoscopoletin suppressed the activation of signaling molecules by 33.24 ± 0.19% for p-PKC, 36.38 ± 2.04% for p-JNK, 81.98 ± 3.15% for c-Fos, and 28.35 ± 1.52% for c-Jun, compared to the only-PI-stimulated control group (Figure 7A,B).

## 3. Discussion

We investigated the effects of isoscopoletin on various immune cells involved in AD and experimentally analyzed the associated mediators. The findings of this analysis suggest it has great potential to identify compounds that could suppress the skin’s inflammatory response. Compounds characterized by relatively high bioavailability and low toxicity are suitable for use as treatments for chronic inflammatory diseases with fewer side effects. In this study, we confirmed that isoscopoletin, found in plants with pharmacological properties, exhibits inhibitory activity on inflammatory mediators involved in AD and its associated mechanisms in HaCaT and RBL-2H3 cells.

In the cell viability measurements using the MTT assay, isoscopoletin showed no significant difference compared to the negative control group for both HaCaT and RBL-2H3 cells up to 40 μM (Figure 2). These results suggest a concentration range of isoscopoletin that can be evaluated for physiological activity without affecting cell viability.

A chemokine is a type of cytokine that recruits various immune cells to tissues. TARC and MDC are chemokines found at particularly high levels in patients with AD, and they trigger skin inflammatory responses by inducing Th2 cell infiltration into the skin [8]. In addition to TARC and MDC, MCP-1 has also been identified as a homeostatic and inflammatory chemokine, which is increased in AD and can support immune cells in skin tissues [43]. IL-8 is an inflammatory chemokine secreted by various cells, including keratinocytes; it induces neutrophil chemotaxis, bactericidal properties, and keratinocyte proliferation [44]. Because these inflammatory mediators are involved in the regulation of skin inflammatory responses, we investigated whether isoscopoletin shows inhibitory effects on them. The results of our analysis confirmed that isoscopoletin inhibits TARC, MDC, MCP-1, IL-8, and IL-1β induction by TI at the protein and mRNA levels in HaCaT cells (Figure 3 and Figure 4).

Cytokines and chemokines are regulated through MAPK and NF-κB, which are important signaling pathways. MAPK participates in various cell responses, including gene expression, cell proliferation, apoptosis, and cell survival. MAPK includes the major subfamilies ERK, JNK, and p38, and the activation of MAPK by TI induces an inflammatory response through the phosphorylation of ERK, JNK, and p38. The activation of NF-κB by TI leads to the translocation of p65 into the nucleus via IκBα degradation, resulting in the expression of inflammatory genes [14,15,17,45]. Therefore, we investigated whether these transcription factors affect the levels of cytokines and chemokines in HaCaT cells. The results revealed that the phosphorylation of MAPKs (ERK, JNK, and p38), NF-κB p65, and IκBα were increased by TI in HaCaT cells, and isoscopoletin inhibited this response (Figure 5).

The JAK/STAT and PI3K/AKT pathways induced by IFN-γ produce various inflammatory cytokines, including TARC [14,15,18]. Although the IFN-γ receptor does not have direct enzymatic activity, its signaling pathway is dependent on JAK, a specific cytoplasmic kinase. When IFN-γ binds to its cognate receptor, JAK is phosphorylated and regulates STAT. When STAT1 and STAT3, family members of STAT, are phosphorylated in TI-treated keratinocytes, they translocate to the nucleus and accumulate, thereby regulating gene expression [15,46]. In the case of the PI3K/AKT pathway, activation by IFN-γ-mediated signaling can contribute to inflammation and the translation of effector proteins [47]. Based on these previous results, since isoscopoletin inhibited MAPK and NF-κB phosphorylation in TI-treated HaCaT cells, we investigated whether isoscopoletin also affects the IFN-γ-mediated signaling pathway. The findings showed that the phosphorylation of STAT1, STAT3, and AKT was increased by TI in HaCaT cells, and isoscopoletin inhibited it (Figure 6). Therefore, these results suggest that isoscopoletin downregulates inflammatory cytokines and chemokines by inhibiting the activation of the MAPK, NF-κB, STAT, and AKT pathways induced by TI in HaCaT cells (Figure 8A).

Patients with AD have a predominant immune response system against Th2 cells in the circulation and body. The potent inducer of Th2-type cytokines, IL-4, is produced by “early” skin-infiltrating T cells. However, it can also be produced by mast cells, eosinophils, and basophils in acute lesions [4]. IL-4 mediates B cell IgE switching and induces IgE production in B cells. Basophils are sensitized by the IgE produced in B cells and migrate to tissues. Then, the basophils release pruritogenic mediators through degranulation and induce cytokines, such as IL-4 and IL-13 [4,6]. IL-4 not only activates Th2 cells but also stimulates sensory neurons to cause itching [27]. Therefore, the regulation of Th2-type cytokines can reduce itching and can improve the destruction of the skin barrier and imbalance between Th1 and Th2 cells through the regulation of Th2 cell differentiation. When the effect of isoscopoletin on IL-4 release was investigated, the results confirmed that isoscopoletin inhibited the protein level of IL-4 induced by PI in RBL-2H3 cells (Figure 3F).

PMA, a PKC activator, and ionomycin a calcium ionophore increasing Ca^2+^ influx, activate PKC. Additionally, activated PKC plays an important role in the allergic response of basophils, leading to the release of inflammatory mediators such as IL-4 [34]. In addition, the phosphorylation of PKC is followed by the activation of the downstream signaling pathway, MAPK and AP-1 [48]. Therefore, the PKC pathway may affect the level of IL-4 regulated by PI in RBL-2H3 cells. In our investigation, PKC, MAPK, and AP-1 were activated by PI in RBL-2H3 cells, and isoscopoletin suppressed their levels (Figure 7). This suggests that isoscopoletin inhibits IL-4 by downregulating the activation of the PKC, MAPK (JNK), and AP-1 pathways induced by PI in RBL-2H3 cells (Figure 8B).

These results suggest that isoscopoletin may be a potential therapeutic candidate for skin diseases, such as AD, by targeting keratinocytes and basophils. In order for isoscopoletin to be used as a direct AD treatment, further research is required on other signaling pathways and in AD animal models. Additionally, because RBL-2H3 cells are derived from rats, we plan to investigate the effects of isoscopoletin on the human-derived KU812 basophil cell line or HMC-1 mast cell line. PMA and ionomycin are non-physiological and artificial inducers, and they activate relatively downstream signaling pathways compared to cytokines. Therefore, further experiments will apply cytokines to support the explanation of other signaling pathways. This study provides a method to identify potential treatments for skin diseases at the in vitro level through these signaling pathways in both HaCaT and RBL-2H3 cell lines.

## 4. Materials and Methods

### 4.1. Materials

Isoscopoletin was obtained from Biosynth (Cat. No. # FI67110, Staad, Switzerland) and dissolved in DMSO. The human keratinocyte cell line HaCaT was obtained from CLS (300493, Eppelheim, Germany). The rat basophil cell line RBL-2H3 was obtained from ATCC (CRL-2256, Manassas, VA, USA). The inducers of HaCaT cells consisted of 10 ng/mL TNF-α (PeproTech, 300-01A, Cranbury, NJ, USA) and 10 ng/mL IFN-γ (PeproTech, 300-02, NJ, USA) and were denoted as TI. The inducers of RBL-2H3 cells consisted of 10 nM PMA (Sigma-Aldrich, P1585, St. Louis, MO, USA) and 100 ng/mL ionomycin (Sigma-Aldrich, 19657, St. Louis, MO, USA) and were denoted as PI. ELISA kits were purchased from R&D systems (TARC set, DY364; MDC set, DY336; MCP-1 set, DY279; IL-8 set, DY208, Minneapolis, MN, USA) and BD (IL-1β set, 557953, East Rutherford, NJ, USA). Primary antibodies against p-ERK (9101), ERK (9102), p-JNK (4668), JNK (9252), p-p38 (9211), p-p65 (3033), p65 (8242), p-STAT1 (9167), p-STAT3 (9145), p-AKT (4060), AKT (4691), p-PKC (2261), c-Fos (2250), and c-Jun (9165) were purchased from Cell Signaling (Danvers, MA, USA). Primary antibodies against p-IκBα (MA5-15087) and IκBα (MA5-15132) were purchased from Invitrogen (Waltham, MA, USA). Primary antibodies against p38 (sc-7972), STAT1 (sc-464), STAT3 (sc-8019), and β-actin (sc-376248) were purchased from Santa Cruz (Dallas, TX, USA).

### 4.2. Cell Culture

HaCaT cells were grown in Dulbecco’s Modified Eagle’s Medium (DMEM; WELGENE, LM001-05, Gyeongsangbuk-do, Republic of Korea) supplemented with heat-inactivated 10% fetal bovine serum (FBS; Gibco, 16000-044, Grand Island, NY, USA) and 1% antibiotics (100 U/mL penicillin, 100 μg/mL streptomycin) and cultured at a temperature of 37 °C in a humidified 5% CO_2_ atmosphere incubator while maintaining a cell confluence of 60–80%. RBL-2H3 cells were grown in Minimum Essential Medium Eagle (MEM; WELGENE, LM007-07, Gyeongsangbuk-do, Republic of Korea), supplemented with heat-inactivated 10% FBS and 1% antibiotics (100 U/mL penicillin, 100 μg/mL streptomycin), and cultured at a temperature of 37 °C in a humidified 5% CO_2_ atmosphere incubator, while maintaining a cell confluence of 60–80%. During subculturing, the cells were detached using trypsin-EDTA (0.05%, 1×) (WELGENE, LS015-01, Gyeongsangbuk-do, Republic of Korea) every 3 days on a 100 mm cell culture dish (CORNING, 353003, Somerville, MA, USA), counted using a hemacytometer (MARIENFELD, 0640010, Lauda-Königshofen, Germany), and placed into new dishes or plates for biological analysis.

### 4.3. Cell Viability Assay

In order to confirm the effect of isoscopoletin on viable cells, the cell viability was measured using the 3-(4,5-dimethylthiazol-2-yl)-2,5-diphenyltetrazolium bromide (MTT; VWR, 0793-1G, West Chester, PA, USA) assay. Cells were seeded in a 96-well cell culture plate (SPL, 30096, Gyeonggi-do, Republic of Korea) at a density of 1 × 10^4^ cells/well. After 4 h of incubation for attaching, isoscopoletin (5, 10, 20, 40, 80, and 160 μM) was added. After 20 h of incubation, the MTT solution (250 μg/mL) was added to each well. After forming formazan for 4 h, the supernatant was aspirated and formazan crystals were dissolved through treatment with 100 μL/well dimethyl sulfoxide (DMSO; SAMCHUN, 000D0458, Seoul, Republic of Korea). Using a Multi-Functional Microplate Reader (TECAN, Spark™ 10M, Mennedorf, Switzerland), the optical density of each well was measured at 570 nm and 620 nm. The formazan value of the experimental group treated with 0.1% DMSO without sample treatment was used as a negative control at 100%.

### 4.4. Measurement of Cytokine and Chemokine Secretion

The cytokine and chemokine secretion levels were analyzed using an enzyme-linked immunosorbent assay (ELISA). For the production of IL-8, MCP-1, and IL-4, HaCaT and RBL-2H3 cells were seeded in a 96-well cell culture plate at a density of 5 × 10^4^ cells/well. For the production of MDC, TARC, and IL-1β, HaCaT cells were seeded in a 12-well cell culture plate (SPL, 30012, Gyeonggi-do, Republic of Korea) at a density of 2.5 × 10^5^ cells/well. After 6 h of incubation to reach confluence, the growth medium was replaced with 1% FBS medium and left overnight. Isoscopoletin (5, 10, 20, and 40 μM) was pretreated for 1 h, and then the cells were stimulated with TI or PI. The next day, the supernatant was harvested. The inhibitory effect of isoscopoletin on cytokines and chemokines in the supernatant was detected using ELISA kits. Using a Multi-Functional Microplate Reader, the optical density of each well was measured at 450 nm and 570 nm. The negative control was set as an experimental group treated with only 0.1% DMSO.

### 4.5. Reverse Transcription-Quantitative PCR (RT-qPCR)

The gene expression levels of the cytokines and chemokines were analyzed using RT-qPCR. HaCaT cells were cultured overnight in a 12-well cell culture plate at a density of 2.5 × 10^5^ cells/well. Subsequently, the growth medium was replaced with 1% FBS medium and the cells were incubated for 2 h. Isoscopoletin (5, 10, 20, and 40 μM) was pretreated for 1 h, and the cells were stimulated with TI for 6 h. After removing the supernatant, the total RNA was isolated by treating the cells with 400 μL/well of XENOSEPA™-TR (XENOHELIX, 93667873-TR, Incheon, Republic of Korea). The RNA was quantified using a UV/Vis Spectrophotometer (Thermo Fisher, NanoDrop™ One, Waltham, MA, USA) and transcribed to cDNA using ReverTra Ace™ qPCR RT Master Mix with gDNA Remover (TOYOBO, FSQ-301, Osaka, Japan). According to the manufacturer’s instructions, as a step to inhibit RNase, an RNA sample of 1 μg, nucleus-free water and 4× RT Master Mix with gDNA Remover were mixed and incubated at 37 °C for 5 min. As a second step for reverse transcription, 5× RT Master Mix was subsequently mixed and incubated at 37 °C for 15 min, 50 °C for 5 min, and 98 °C for 5 min. The synthesized cDNA was mixed with primer and KAPA SYBR^®^ FAST (KAPA BIOSYSTEMS, KK4602, Wilmington, MA, USA), and RT-qPCR was performed on a QuantStudio™ 1 Real Time PCR system (Thermo Fisher, A40425, Waltham, MA, USA). The relative mRNA level of the target gene was calculated using the ^ΔΔ^C*t* method and normalized to glyceraldehyde 3-phosphate dehydrogenase (GAPDH), which was used as the housekeeping gene. The primer sequences used are listed below (Table 1). The negative control was set as an experimental group treated with only 0.1% DMSO.

### 4.6. Western Blot Analysis

The transcription factor protein levels were analyzed using sodium dodecyl sulfate-polyacrylamide gel electrophoresis (SDS-PAGE) and Western blot. Cells were cultured overnight in a 60 mm cell culture dish (Eppendorf, 0030701119, Hamburg, Germany) at a density of 2 × 10^6^ cells/dish. Then, the growth medium was replaced with 1% FBS medium and the cells were incubated for 2 h. Isoscopoletin (5, 10, 20, and 40 μM) was pretreated for 1 h, and the cells were stimulated with TI or PI at different times for each target. The negative control was set as an experimental group treated with only 0.1% DMSO. After removing the supernatant, the cells were washed with 1× phosphate-buffered saline (PBS; Tech & Innovation, BPB-9121, Gangwon-do, Republic of Korea). Subsequently, the cells were lysed with 100 μL lysis solution (20 mM Tris-HCl (pH 7.4), 50 mM NaCl, 50 mM Na pyrophosphate, 30 mM NaF, 5 μM zinc chloride, 2 mM iodoacetic acid, and 1% Triton^®^ X-100 detergent (BIO-RAD, 161-0407, Hercules, CA, USA) in distilled water supplemented with 1 mM phenylmethylsulfonyl fluoride (PMSF) protease inhibitor (Thermo Fisher, 36978, Waltham, MA, USA) and 0.1 mM sodium orthovanadate (Sigma-Aldrich, 567540, St. Louis, MO, USA), and the harvested lysate was centrifuged at 13,000× *g* RPM for 5 min at 4 °C. The total protein concentration of the supernatant was quantified using the Pierce™ BCA protein assay kit (Thermo Fisher, 23227, Waltham, MA, USA). Equal amounts of the protein samples were separated using SDS-PAGE and transferred to an Immobilon^®^-P polyvinylidene fluoride (PVDF) membrane (MERCK, IPVH00010, Darmstadt, Germany). To prevent nonspecific binding, the membrane was blocked with 5% Difco™ skim milk (BD, 232100, East Rutherford, NJ, USA) in 1× Tris-buffered saline with Tween 20 (TBS-T; LPS solution, CBT007, Daejeon, Republic of Korea) for 1 h at room temperature (RT). After washing 3 times with 1× TBS-T, the membrane was reacted with a primary antibody diluted in a 5% skim milk solution overnight at 4 °C. After a subsequent washing 3 more times with 1× TBS-T, the membrane was reacted with horseradish peroxidase (HRP)-conjugated secondary antibodies diluted in a 5% skim milk solution for 1 h at RT. The visualization of the membrane was achieved using a luminescent image analyzer (General Electric, Amersham™ Imager 680, Boston, MA, USA) by exposing it to a Pierce™ enhanced chemiluminescence (ECL) Western blotting substrate (Thermo Fisher, 23227, Waltham, MA, USA) that specifically binds to HRP. The densitometry of the protein bands was quantified using ImageJ software (National Institutes of Health, version 1.52a, Stapleton, NY, USA), and normalized β-actin was used as the housekeeping gene.

### 4.7. Immunocytochemistry (ICC) Analysis

HaCaT cells were cultured overnight in a Nunc™ Lab-Tek™ II Chamber Slide™ system (Thermo Fisher, 154534, Waltham, MA, USA) at a density of 2 × 10^4^ cells/well. Then, the growth medium was replaced with 1% FBS medium and the cells were incubated for 2 h. Isoscopoletin (5, 10, 20, and 40 μM) was pretreated for 1 h, and the cells were stimulated with TI for 1 h. After washing 3 times with 1× PBS, the cells were fixed in 4% formaldehyde solution (Sigma-Aldrich, 252549, St. Louis, MO, USA) in 1× PBS for 3 min at RT. After washing 3 times with 1× PBS, to increase membrane permeability, the cells were treated with 0.3% Triton^®^ X-100 detergent in 1× PBS for 10 min at 4 °C. After washing 3 times with 1× PBS, the fixed cells were blocked with a 3% bovine serum albumin (BSA; Gibco, 30063-572, Grand Island, NY, USA) for 1 h at 4 °C. After washing 3 times with 1× PBS, the fixed cells were reacted with primary antibody NF-κB p65 diluted in a 3% BSA solution overnight at 4 °C. After washing 5 times with 1× PBS, the fixed cells were reacted with Alexa Fluor™ 488-conjugated secondary antibody (Invitrogen, A21202, Waltham, MA, USA) diluted in a 3% BSA solution for 1 h at RT in the dark. After washing 5 times with 1× PBS, Hoechst 33342 (Invitrogen, H3570, Waltham, MA, USA) was used to treat the fixed cells to stain the nuclei for 5 min at RT in the dark. After washing 5 times with 1× PBS, the cells were mounted using VECTASHIELD^®^ mounting medium (Vectorlabs, H-1000, Newark, CA, USA). The fixed cells were observed and photographed using a microscope (Axio Observer Z1, Carl ZEISS, Oberkochen, Germany).

### 4.8. Statistical Analysis

All data are expressed as the mean standard deviation (SD) and compared between two independent experiments. Statistical significance between groups was determined using Student’s *t*-test, where values of *p* < 0.05 were considered to indicate a statistically significant difference.

## 5. Conclusions

Isoscopoletin has no significant effect on cell viability at concentrations below 40 μM in either HaCaT or RBL-2H3 cells. In TI-stimulated HaCaT cells, isoscopoletin inhibited the release of TARC, MDC, MCP-1, IL-8, and IL-1β. Moreover, the phosphorylation of MAPK, NF-κB, STAT, and AKT was increased by TI, but isoscopoletin suppressed these transcription factors. In PI-stimulated RBL-2H3 cells, isoscopoletin inhibited the release of IL-4. Moreover, the activation of PKC, MAPK, and AP-1 was increased by PI, but isoscopoletin suppressed these transcription factors. These results demonstrate that isoscopoletin ameliorates inflammation by regulating upstream signaling pathways in TI-stimulated HaCaT and PI-stimulated RBL-2H3 cells. Therefore, we suggest that isoscopoletin may have potential as a functional treatment for skin inflammation, such as that found in AD, due to its anti-inflammatory and anti-allergic effects.

## Figures and Tables

**Figure 1 ijms-25-06908-f001:**
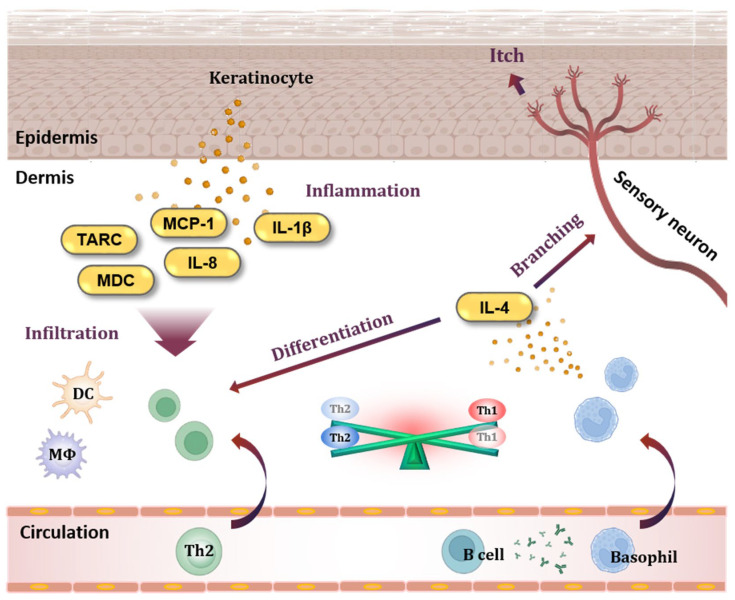
Keratinocytes and basophils in AD pathogenesis. When keratinocytes in the epidermis are activated, they release chemokines such as TARC/CCL17, MDC/CCL22, MCP-1/CCL2, and IL-8/CXCL8, which are specifically increased in AD. These chemokines cause immune cells to infiltrate tissues, and cytokines such as IL-1β cause inflammation in the skin. In addition, antibodies produced from B cells sensitize basophils. Basophils migrate to tissues and release IL-4, a Th2-type cytokine, causing an imbalance between Th2 cells and Th1 cells. In addition, IL-4 disrupts the skin barrier by stimulating sensory neurons. TARC, thymus- and activation-regulated chemokine; MDC, macrophage-derived chemokine; MCP-1, monocyte chemoattractant protein-1; IL-8, interleukin-8; IL-1β, interleukin-1beta; DC, dendritic cell; MΦ, macrophage; Th1, T helper type 1 cell; Th2, T helper type 2 cell; IL-4, interleukin-4.

**Figure 2 ijms-25-06908-f002:**
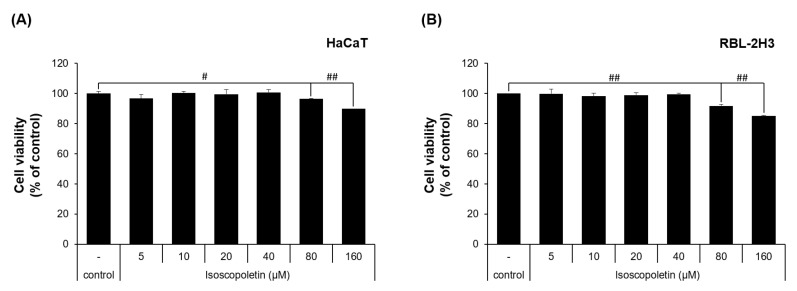
Viability of HaCaT and RBL-2H3 cells treated with isoscopoletin. (**A**,**B**) Cells were seeded in a 96-well plate at a density of 1 × 10^4^ cells/well. After 4 h, cells were treated with isoscopoletin (5, 10, 20, 40, 80, and 160 μM) for 20 h. Cell viability was measured using MTT assay. The formazan value of the experimental group treated only with 0.1% DMSO was used as a negative control and set at 100%. Data are the mean ± SD, and statistical significance was assessed via *t*-test. # *p* < 0.05 and ## *p* < 0.01 versus negative control group.

**Figure 3 ijms-25-06908-f003:**
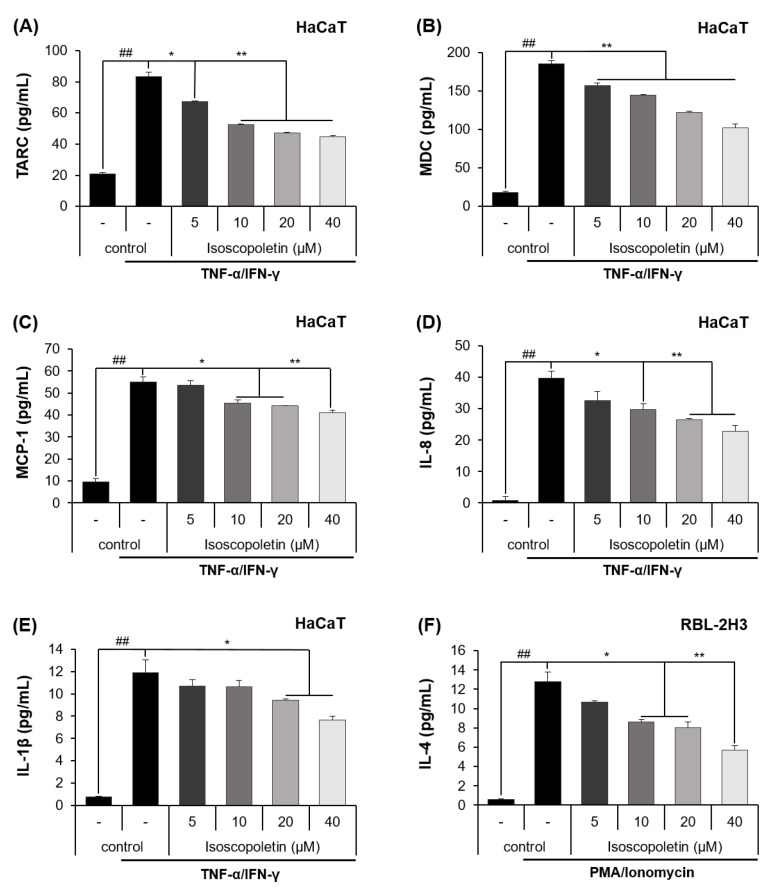
Effect of isoscopoletin on the secretion of cytokines and chemokines in TI-stimulated HaCaT and PI-stimulated RBL-2H3 cells. Cells were seeded in a 96-well cell culture plate at a density of 2 × 10^4^ cells/well. After 6 h, cells were treated with isoscopoletin (5, 10, 20, and 40 μM) for 1 h and TI or PI overnight. The secretion levels of (**A**) TARC, (**B**) MDC, (**C**) MCP-1, (**D**) IL-8, (**E**) IL-1β, and (**F**) IL-4 in the supernatant were detected using ELISA kits. Data are the mean ± SD, and statistical significance was assessed via *t*-test. ## *p* < 0.01 versus negative control group; * *p* < 0.05 and ** *p* < 0.01 versus TI or PI-only group. The different colored bars represent variations according to the experimental groups. TI, TNF-α/IFN-γ; PI, PMA/ionomycin.

**Figure 4 ijms-25-06908-f004:**
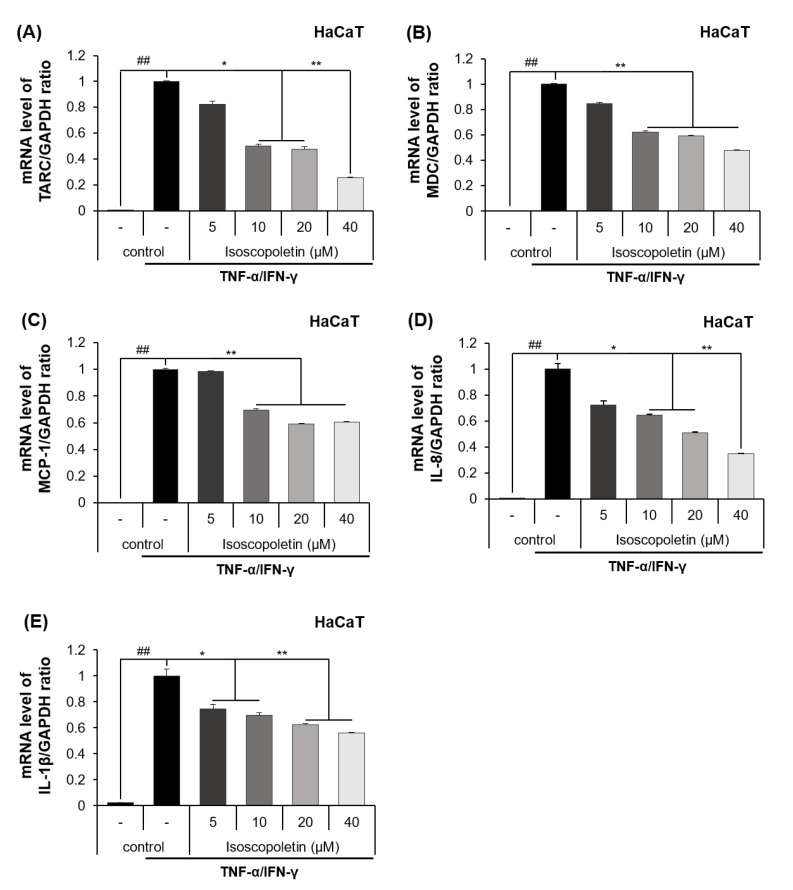
Effect of isoscopoletin on the mRNA expression of cytokines and chemokines in TI-stimulated HaCaT cells. HaCaT cells were seeded in a 12-well cell culture plate at a density of 2.5 × 10^5^ cells/well overnight. Cells were treated with isoscopoletin (5, 10, 20, and 40 μM) for 1 h and TI for 6 h. The mRNA levels of (**A**) TARC, (**B**) MDC, (**C**) MCP-1, (**D**) IL-8, and (**E**) IL-1β were detected using RT-qPCR. Data are the mean ± SD, and statistical significance was assessed via *t*-test. ## *p* < 0.01 versus negative control group; * *p* < 0.05 and ** *p* < 0.01 versus TI-only group. The different colored bars represent variations according to the experimental groups. TI, TNF-α/IFN-γ.

**Figure 5 ijms-25-06908-f005:**
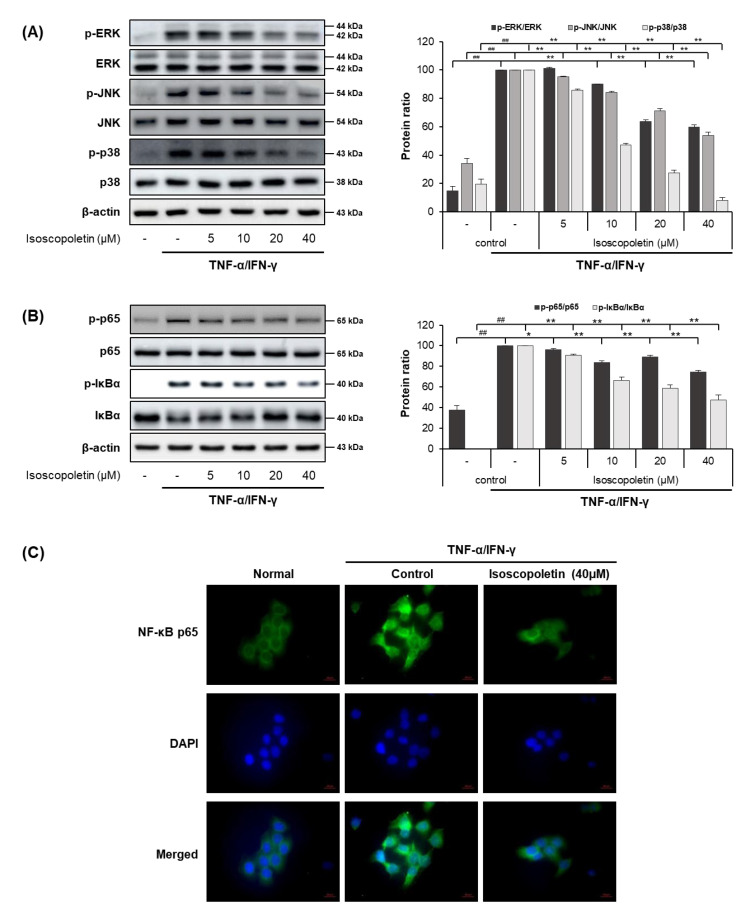
Effect of isoscopoletin on MAPK and NF-κB in TI-stimulated HaCaT cells. HaCaT cells were pretreated with isoscopoletin for 1 h, and proteins were extracted after TI treatment for (**A**) 30 min (MAPK) and (**B**) 1 h (NF-κB). Proteins were analyzed via Western blot using specific antibodies. (**C**) Translocation of NF-κB p65 was detected using ICC, scale bar = 20 µm. Data are the mean ± SD, and statistical significance was assessed via *t*-test. ## *p* < 0.01 versus negative control group; * *p* < 0.05 and ** *p* < 0.01 versus TI-only group. The different colored bars represent variations according to the experimental groups. TI, TNF-α/IFN-γ.

**Figure 6 ijms-25-06908-f006:**
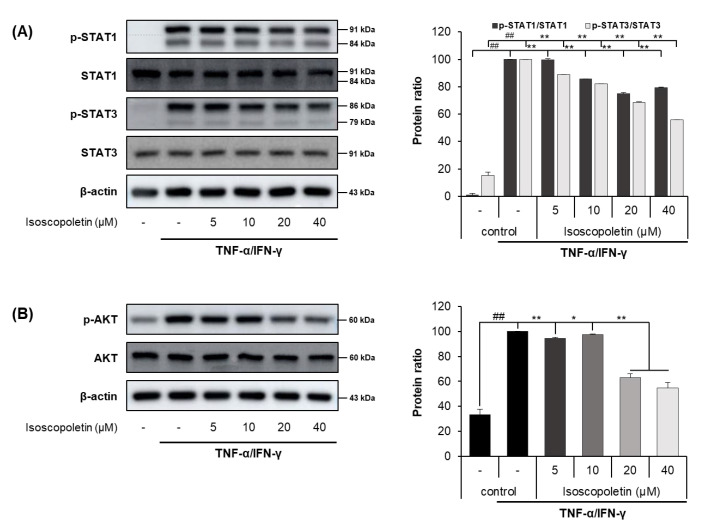
Effect of isoscopoletin on STAT and AKT in TI-stimulated HaCaT cells. HaCaT cells were pretreated with isoscopoletin for 1 h, and proteins were extracted after TI treatment for (**A**) 1 h (AKT) and (**B**) 2 h (STAT). Proteins were analyzed via Western blot using specific antibodies. Data are the mean ± SD, and statistical significance was assessed via *t*-test. ## *p* < 0.01 versus negative control group; * *p* < 0.05 and ** *p* < 0.01 versus TI-only group. The different colored bars represent variations according to the experimental groups. TI, TNF-α/IFN-γ.

**Figure 7 ijms-25-06908-f007:**
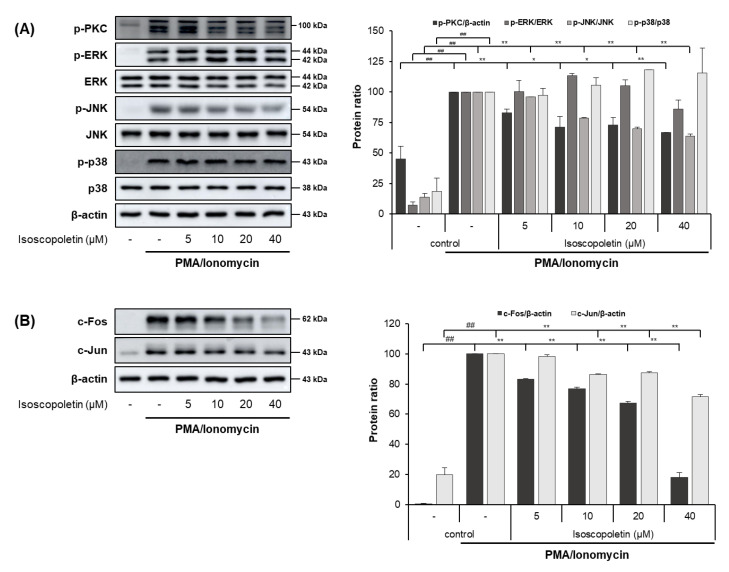
Effect of isoscopoletin on PKC, MAPK, and AP-1 in PI-stimulated RBL-2H3 cells. RBL-2H3 cells were pretreated with isoscopoletin for 1 h, and proteins were extracted after PI treatment for (**A**) 30 min (PKC and MAPK) and (**B**) 1 h (AP-1). Proteins were analyzed via Western blot using specific antibodies. Data are the mean ± SD, and statistical significance was assessed via *t*-test. ## *p* < 0.01 versus negative control group; * *p* < 0.05 and ** *p* < 0.01 versus PI-only group. The different colored bars represent variations according to the experimental groups. PI, PMA/ionomycin.

**Figure 8 ijms-25-06908-f008:**
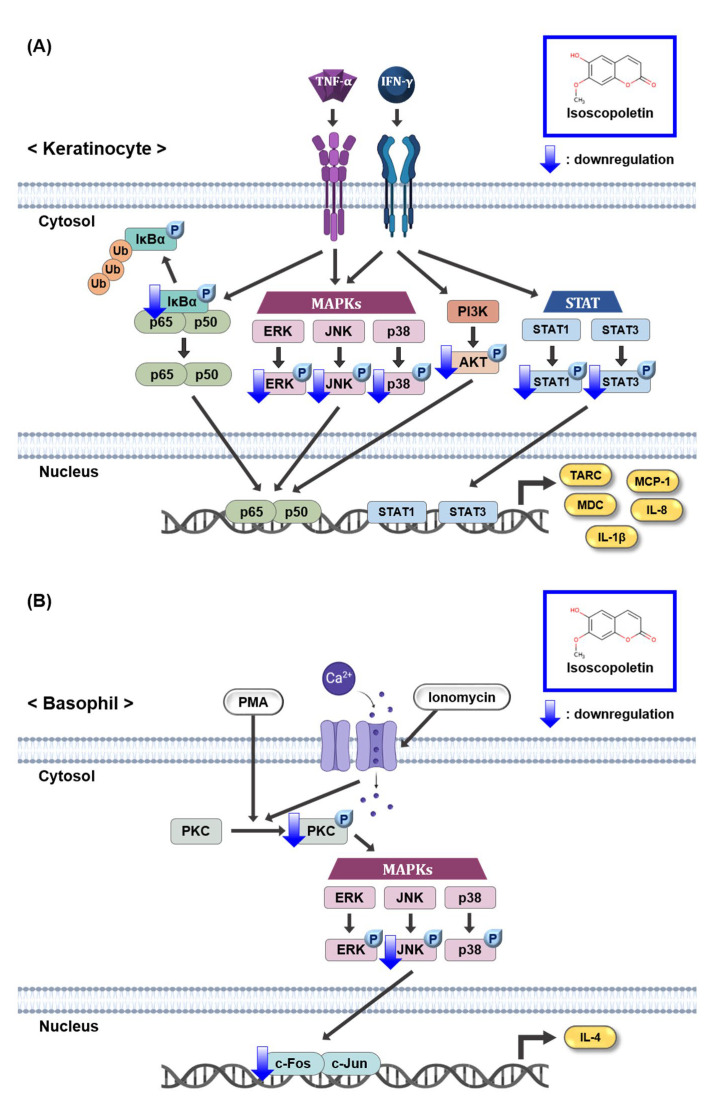
Schematic mechanism showing the effect of isoscopoletin in TI-stimulated HaCaT and PI-stimulated RBL-2H3 cells. (**A**) Isoscopoletin downregulates cytokines and chemokines by inhibiting the activation of MAPK, NF-κB, STAT, and AKT pathways induced by TI in HaCaT cells. (**B**) Isoscopoletin downregulates IL-4 by inhibiting the activation of PKC, MAPK (JNK), and AP-1 pathways induced by PI in RBL-2H3 cells. TNF-α, tumor necrosis factor-alpha; IFN-γ, interferon-gamma; TI, TNF-α/IFN-γ; NF-κB, nuclear factor kappa-light-chain-enhancer of activated B cells; IκBα, NF-κB inhibitor, alpha; Ub, ubiquitination; MAPK, mitogen-activated protein kinase; ERK, extracellular signal-regulated kinase; JNK, c-Jun N-terminal kinase; PI3K, phosphoinositide 3-kinase; AKT, protein kinase B (PKB); STAT1, signal transducer and activator of transcription 1; STAT3, signal transducer and activator of transcription 3; TARC, thymus- and activation-regulated chemokine; MDC, macrophage-derived chemokine; MCP-1, monocyte chemoattractant protein-1; IL-8, interleukin-8; IL-1β, interleukin-1beta; PMA, phorbol 12-myristate 13-acetate; PI, PMA/ionomycin; PKC, protein kinase C; IL-4, interleukin-4.

**Table 1 ijms-25-06908-t001:** Primer sequences used in this study.

Cell Line	Gene	Direction	Sequence (5′→3′)
Human HaCaT	TARC	Forward	CAC GCA GCT CGA GGG ACC AAT GTG
		Reverse	TCA AGA CCT CTC AAG GCT TTG CAG G
	MDC	Forward	AGG ACA GAG CAT GGC TCG CCT ACA GA
		Reverse	TAA TGG CAG GGA GGT AGG GCT CCT GA
	MCP-1	Forward	TCT GTG CCT GCT GCT CAT AG
		Reverse	CAG ATC TCC TTG GCC ACA AT
	IL-8	Forward	ATG ACT TCC AAG CTG GCC GTG GCT
		Reverse	TTA TGA ATT CTC AGC CCT CTT CAA AAA
	IL-1β	Forward	CTG TCG TGC GTG TTG AAA GA
		Reverse	TTC TGC TTG AGA GGT GCT GA
	GAPDH	Forward	CGG AGT CAA CGG ATT TGG TCG
		Reverse	AGC CTT CTC CAT GGT GGT GAA G

## Data Availability

The data presented in this study are available upon request from the corresponding author.

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
