# Peer review of "Effect of Isoscopoletin on Cytokine Expression in HaCaT Keratinocytes and RBL-2H3 Basophils: Preliminary Study"

_ijms, 2024, doi:10.3390/ijms25136908_

Round 1
Reviewer 1 Report
Comments and Suggestions for Authors
This concise in vitro study is in attempt to investigate the anti-inflammatory activities of natural compound isoscopoletin on the atopic dermatitis-mediated inflammation using TNF-α/IFN-γ-treated HaCaT keratinocyte cell line and PMA/ionomycin treated RBL-2H3 basophil cell line. Results showed that isoscopoletin could reduce the TARC/CCL17, MDC/CCL22, MCP-1/CCL2, IL-8/CXCL8 and IL-1β in TNF-α/IFN-γ-treated HaCaT cells and IL-4 in PMA/ionomycin-treated RBL-2H3 cells, by regulating the intracellular MAPK, NF-κB, STAT and AKT/PKB, and PKC, MAPK and AP-1, respectively.
Major comments
(1) The major concerns are the use of cell lines instead of the primary human keratinocytes and primary human basophils for the in vitro experiments. Alternatively, human KU812 basophilic cells should be better than rat basophilic cell line RBL-2H3 cells for the basophil experiment.
(2) Apart from basophils, eosinophils also play important roles in allergic inflammation in AD. Therefore, eosinophils should also be investigated in this study.
(3) Since the intercellular interaction is important for allergic inflammation in AD, the co-culture experiments by using keratinocytes together with basophils should be also be investigated to confirm the anti-inflammatory activity of isoscopoletin.
(4) Since PMA and ionomycin are not physiological stimulators of basophils, more relevant basophil activating cytokines including IL-3, IL-31 and IL-33 should also be used for the basophil activation.
(5) Figure 4 should also include the mRNA expression of IL-4 in PMA/ionomycin-treated RBL-2H3 cells.
Minor comments
(1) The chemical structure of isoscopoletin should be shown. What is the solvent (e.g. methanol, ethanol, DMSO) used to dissolve the isoscopoletin?
(2) In line 150 “using biomarkers that can identify autoimmune disease”, “autoimmune disease” may be “allergic disease”. Please check.
Author Response
Reviewer 1.
[Comments and Suggestions for Authors]
This concise in vitro study is in attempt to investigate the anti-inflammatory activities of natural compound isoscopoletin on the atopic dermatitis-mediated inflammation using TNF-α/IFN-γ-treated HaCaT keratinocyte cell line and PMA/ionomycin treated RBL-2H3 basophil cell line. Results showed that isoscopoletin could reduce the TARC/CCL17, MDC/CCL22, MCP-1/CCL2, IL-8/CXCL8 and IL-1β in TNF-α/IFN-γ-treated HaCaT cells and IL-4 in PMA/ionomycin-treated RBL-2H3 cells, by regulating the intracellular MAPK, NF-κB, STAT and AKT/PKB, and PKC, MAPK and AP-1, respectively.
Major comments
(1) The major concerns are the use of cell lines instead of the primary human keratinocytes and primary human basophils for the in vitro experiments. Alternatively, human KU812 basophilic cells should be better than rat basophilic cell line RBL-2H3 cells for the basophil experiment.
=> Thank you for your valuable comments and suggestions.
To set up an in vitro system for developing a disease treatment, experimental conditions must be set as similar as possible to the disease. Therefore, it may be experimentally appropriate to use primary cells derived from tissue of AD patients. However, they are more sensitive and difficult to handle than immortalized cell lines, so cell lines are generally easier to be used in experiments and many references exist.
The keratinocyte HaCaT cell line is mainly used as an in vitro model of AD. Given that AD is a disease exacerbated by persistent itching, it is also essential to study cell lines that serve as markers for itching. Mast cells and basophils release IL-4 that stimulates nerve neurons, and the RBL-2H3 cell line exhibits characteristics of both mast cells and basophils.
We have planned the experiments by comparing studies that used the HaCaT cell line alone and papers that used various cell lines. (Int J Mol Sci. 2022 Nov 24;23(23):14642. doi: 10.3390/ijms232314642., Int J Mol Sci. 2023 Aug 15;24(16):12803. doi: 10.3390/ijms241612803., Int J Mol Sci. 2022 Aug 17;23(16):9279. doi: 10.3390/ijms23169279., Int J Mol Sci. 2022 Oct 11;23(20):12072. doi: 10.3390/ijms232012072., Molecules. 2019 Jan 11;24(2):265. doi: 10.3390/molecules24020265.)
Accroding to your comments, we added the further study in line 322, in the Discussion section.
Additionally, because RBL-2H3 cells are derived from rats, we plan to investigate the effects of isoscopoletin on the human-derived KU812 basophil cell line or HMC-1 mast cell line. PMA and ionomycin are non-physiological and artificial inducers, and they activate relatively downstream signaling pathways compared to cytokines. Therefore, further experiments will apply cytokines to support the explanation of other signaling pathways. This study provides a method to identify potential treatments for skin diseases at the in vitro level through these signaling pathways in both HaCaT and RBL-2H3 cell lines.
(2) Apart from basophils, eosinophils also play important roles in allergic inflammation in AD. Therefore, eosinophils should also be investigated in this study.
=> Thank you for your valuable suggestions.
Eosinophils, a type of granulocyte, are involved in parasitic defense and allergic reactions (Nat Rev Immunol. 2017 Dec;17(12):746-760. doi: 10.1038/nri.2017.95.). IL-5, specifically expressed by eosinophils, along with other inflammatory cytokines, can serve as important markers in AD (Front Physiol. 2019 Dec 17;10:1514. doi: 10.3389/fphys.2019.01514.). The significance of this experiment was to improve the suitability of the sample effect to the AD model by simultaneously applying other cells rather than using alone the HaCaT cell line, which is mainly utilized as an in vitro model of AD. Therefore, we plan to use the human-derived EOL-1 cell line in further experiments.
We referred to studies that utilized the HaCaT cell line and cell lines indicating markers of itching (Molecules. 2019 Jan 11;24(2):265. doi: 10.3390/molecules24020265., Molecules. 2020 Mar 28;25(7):1554. doi: 10.3390/molecules25071554.).
(3) Since the intercellular interaction is important for allergic inflammation in AD, the co-culture experiments by using keratinocytes together with basophils should be also be investigated to confirm the anti-inflammatory activity of isoscopoletin.
=> Thank you for your valuable suggestions.
AD disease appears as a result of complex interactions between various cells in the skin environment. Therefore, co-culture may be necessary to design experiments that closely resemble such an environment. However, the primary aim of this experiment was to identify the mechanisms of targeted AD factors and explore the block markers of the samples. Once the signaling pathways influenced by the samples are identified, cell-cell interaction can be explained through additional co-culture experiments. Thus, while futher experiments may consider co-culture studies, we also plan to conduct in vivo experiments using an Atopic Dermatitis Mouse Model.
We referred to studies that evaluated sample activity by analyzing signaling pathways using various cell lines individually (Molecules. 2019 Jan 11;24(2):265. doi: 10.3390/molecules24020265., Molecules. 2021 May 30;26(11):3298. doi: 10.3390/molecules26113298., Int J Mol Sci. 2021 Aug 5;22(16):8431. doi: 10.3390/ijms22168431.).
(4) Since PMA and ionomycin are not physiological stimulators of basophils, more relevant basophil activating cytokines including IL-3, IL-31 and IL-33 should also be used for the basophil activation.
=> Thank you for your valuable comment.
PMA induces IL-13 and ionomycin induces IL-4, but it is known that a synergistic effect occurs when treated together (Clin Exp Immunol. 1997 May;108(2):295-301. doi: 10.1046/j.1365-2249.1997.d01-1001.x.). PMA and ionomycin are non-physiological and artificial inducers, and they activate relatively downstream signaling pathways compared to cytokines. Nevertheless, the reason for using PMA and ionomycin is that their mechanisms of action in RBL-2H3 cells have been previously elucidated, allowing for the rapid targeting of molecules from a signaling pathway perspective (J Agric Food Chem. 2015 Jan 14;63(1):192-9. doi: 10.1021/jf504013n.). If the sample had no effect on the signaling pathways induced by PMA and ionomycin, other inducers could be used. However, since the effect of the sample was confirmed first, further experiments will apply cytokines to support the explanation of other signaling pathways. The limitations of the study were additionally reflected in line 324, in the Discussion section.
“PMA and ionomycin are non-physiological and artificial inducers, and they activate relatively downstream signaling pathways compared to cytokines. Therefore, further experiments will apply cytokines to support the explanation of other signaling pathways.”
(5) Figure 4 should also include the mRNA expression of IL-4 in PMA/ionomycin-treated RBL-2H3 cells.
=> Thank you for your valuable comment.
As in the HaCaT cell line, we detected the mRNA expression levels in the RBL-2H3 cell line. Isoscopoletin significantly reduced the amount of IL-4 in ELISA (Fig. 3F), but no significant inhibitory effect was found at the mRNA level. Nevertheless, because IL-4 is important to verify whether the sample affects the regulation of itching, only ELISA result were presented. In the future, additional experiments are needed to confirm the effects of isoscopoletin on mRNA levels of IL-4 in various PCR condition.
Minor comments
(1) The chemical structure of isoscopoletin should be shown. What is the solvent (e.g. methanol, ethanol, DMSO) used to dissolve the isoscopoletin?
=> Thank you for your valuable comment.
Isoscopoletin was dissolved in DMSO
In line 351, we added the expression “and dissolved in DMSO”.
(2) In line 150 “using biomarkers that can identify autoimmune disease”, “autoimmune disease” may be “allergic disease”. Please check.
=> Thank you for your valuable comment.
In line 105, we changed the expression to “using biomarkers that can identify allergic disease”.

Reviewer 2 Report
Comments and Suggestions for Authors
Isoscopoletin, a plant-derived compound traditionally used for treating skin diseases, has not been yet been studied on atopic dermatitis (AD). Given the side effects of common AD treatments, this study explored the property of isoscopoletin to regulate inflammatory mediators in TNF-α/IFN-γ-treated HaCaT cells and PMA/ionomycin-treated RBL-2H3 cells. Using MTT assays, ELISA, RT-qPCR, Western blots, and ICC, the study found that isoscopoletin did not affect cell viability at concentrations up to 40 μM and suppressed several inflammatory mediators and signaling pathways in both cell types. These findings suggest the potential therapeutic effects of isoscopoletin in AD.
The study is simple but well-executed, with clear and informative figures. The writing style is concise and effective. The methodology is sound, and the results are presented in a logical and coherent manner. Overall, the study contributes to the evaluation of the potential therapeutic effects of isoscopoletin on atopic dermatitis.
Suggestions for improvement:
-Figure 4 could be used as Suppl. Material, as the protein levels of the same mediators have been already presented in Figure 3
-Figure 5-6-7, the labeling “Relative Expression levels” in the histograms is not appropriate, as they show the ratio between phospho and total protein levels. Please correct.
-Statistics: Did the authors check for normality? Did they adjust for multiple comparisons? The statistical section must be improved.
-The paper is limited to an in vitro study. The limitations of the study must be clearly indicated in the Discussion section
Comments on the Quality of English LanguageMinor editing required
Author Response
Thank you for your valuable comments and suggestions.
Reviewer 2.
[Comments and Suggestions for Authors]
Isoscopoletin, a plant-derived compound traditionally used for treating skin diseases, has not been yet been studied on atopic dermatitis (AD). Given the side effects of common AD treatments, this study explored the property of isoscopoletin to regulate inflammatory mediators in TNF-α/IFN-γ-treated HaCaT cells and PMA/ionomycin-treated RBL-2H3 cells. Using MTT assays, ELISA, RT-qPCR, Western blots, and ICC, the study found that isoscopoletin did not affect cell viability at concentrations up to 40 μM and suppressed several inflammatory mediators and signaling pathways in both cell types. These findings suggest the potential therapeutic effects of isoscopoletin in AD.
The study is simple but well-executed, with clear and informative figures. The writing style is concise and effective. The methodology is sound, and the results are presented in a logical and coherent manner. Overall, the study contributes to the evaluation of the potential therapeutic effects of isoscopoletin on atopic dermatitis.
Suggestions for improvement:
-Figure 4 could be used as Suppl. Material, as the protein levels of the same mediators have been already presented in Figure 3
=> Thank you for your valuable suggestion.
The targets identified in Figures 3 and 4 are the same, but they differ depending on whether they are at the protein level or the gene level. The aim of this study is to determine whether the sample affects inflammatory factors and, if so, which signaling pathways. Therefore, it was possible to design an experiment to predict transcription factors through the level of cytokines released from cells, and the gene levels resulting from the transcription factors.
-Figure 5-6-7, the labeling “Relative Expression levels” in the histograms is not appropriate, as they show the ratio between phospho and total protein levels. Please correct.
=> Thank you for your valuable comment.
The expression “relative expression levels” in figure 5-6-7 has been revised to “protein ratio”.
Figure 5. Effect of isoscopoletin on MAPK and NF-κB in TI-stimulated HaCaT cells.
Figure 6. Effect of isoscopoletin on STAT and AKT in TI-stimulated HaCaT cells.
Figure 7. Effect of isoscopoletin on PKC, MAPK, and AP-1 in PI-stimulated RBL-2H3 cells.
-Statistics: Did the authors check for normality? Did they adjust for multiple comparisons? The statistical section must be improved.
=> Thank you for your valuable comment.
The expression “All data are expressed as the mean standard deviation (SD).” in manuscript line 468 has been revised to “All data are expressed as the mean standard deviation (SD) and compared between two independent experiments.”
-The paper is limited to an in vitro study. The limitations of the study must be clearly indicated in the Discussion section.
=> Thank you for your valuable comment.
The limitations of the study were additionally reflected in manuscript line 322 in the discussion section.
“Additionally, because RBL-2H3 cells are derived from rats, we plan to investigate the effects of isoscopoletin on the human-derived KU812 basophil cell line or HMC-1 mast cell line. PMA and ionomycin are non-physiological and artificial inducers, and they activate relatively downstream signaling pathways compared to cytokines. Therefore, further experiments will apply cytokines to support the explanation of other signaling pathways.”

Round 2
Reviewer 1 Report
Comments and Suggestions for Authors
As mentioned, the study needs additional experiments such as the in vitro experiments using primary human keratinocytes and primary human basophils.
Author Response
Comment; As mentioned, the study needs additional experiments such as the in vitro experiments using primary human keratinocytes and primary human basophils.
Response; Thank you for your valuable suggestions.
I agree with you that this study should have been designed more thoroughly designed using human primary cells.
Therefore, since immortalized cell lines were used in this manuscript, it can be considered as a preliminary study, so I revised the title to ”Effect of Isoscopoletin on Cytokine Expression in HaCaT Keratinocytes and RBL-2H3 Basophils: preliminary study”.
I believe that Because primary cells require more stringent conditions for stable cell culture than immortalized cell lines, optimal conditions for the primary cell culture must first be established, which is difficult to achieve in a short period of time. Since I have no experience culturing primary cells, we plan to establish a cell culture system and proceed with this in subsequent studies. As a result, based on the current research, it is premature to conclude the availability of isoscopoletin as an AD treatment. Thus, I propose presenting isoscopoletin as a potential AD candidate substance, and aim to increase its availability in AD treatment by establishing in vitro and in vivo experimental system in further studies.
Additionally, I would like to consider experiments using not only primary cells, but also alternative models such as skin-on-a-chip (Int. J. Mol. Sci. 2022, 23(4), 2116) or KeraSkinTM (Toxicol In Vitro. 2014 Aug;28(5):742-50), which are developed using human cells.
Int. J. Mol. Sci. 2022, 23(4), 2116. An Interleukin-4 and Interleukin-13 Induced Atopic Dermatitis Human Skin Equivalent Model by a Skin-On-A-Chip; https://doi.org/10.3390/ijms23042116
Toxicol In Vitro. 2014 Aug;28(5):742-50. KeraSkin-VM: a novel reconstructed human epidermis model for skin irritation tests. doi: 10.1016/j.tiv.2014.02.014.
Reviewer 2 Report
Comments and Suggestions for Authors
ALL THE ISSUES HAVE BEEN ADDRESSED
Author Response
Thank you for your comment "ALL THE ISSUES HAVE BEEN ADDRESSED".
Round 3
Reviewer 1 Report
Comments and Suggestions for Authors
The manuscript can be accepted in present form.
Comments on the Quality of English LanguageMinor editing of English language required.